# Analysing Multiple Paths of Urban Low-Carbon Governance: A Fuzzy-Set Qualitative Comparative Analysis Method Based on 35 Key Cities in China

You-Dong Li , Chen-Li Yan, Yun-Hui Zhao * and Jia-Qi Bai

School of Business Administration, Inner Mongolia University of Finance and Economics, Hohhot 010051, China
*   Correspondence: zhaoyunhui602@126.com

**Abstract:** The city is a crucial space carrier for the country to carry out low-carbon construction and solve sustainable–development problems. However, existing research lacks an in-depth discussion of the complex mechanisms and governance paths of urban low-carbon transformation. Therefore, this study explores multiple paths of urban low-carbon governance (ULCG). This study constructs a theoretical model of ULCG based on the technology–organisation–environment (TOE) framework. It uses fuzzy-set qualitative comparative analysis (fsQCA) to analyse the overall and sub-regional paths of 35 key cities in China to explore various ULCG approaches. The following three conclusions are drawn. First, a single antecedent condition is not a necessary condition for ULCG. Second, five differentiated paths have been formed under the joint action of the TOE conditions to improve ULCG. It can be divided into three types: the ULCG model dominated by 'big data + market', 'big data', and 'market'. Third, apparent differences exist in the ULCG paths in China's eastern, central and western regions. The study deepens the rational understanding of multiple factors interacting in the complex mechanism behind urban low-carbon transformation and provides differentiated ULCG paths, enabling cities in eastern, central, and western China to choose low-carbon governance paths tailored to their local conditions based on both a comprehensive perspective and a regional perspective.

**Keywords:** ULCG; TOE framework; sub-regional paths; fsQCA

## 1. Introduction

Global carbon dioxide concentration levels reached their peak in 2019, with urban carbon emissions constituting 70–80% of total carbon emissions [1]. The entire operational process of a city consumes considerable fossil energy [2]. As a practitioner and an important participant in ecological civilisation, China considers urban low-carbon development vital. The National Development and Reform Commission announced low-carbon pilot cities in three batches, and each city has set Carbon Peaking and Carbon Neutrality Goals [3]. In the future, low-carbon construction will become imperative when investigating cities' international competitiveness and sustainable development.

How can we actively and effectively promote urban low-carbon governance (ULCG)? At an organisational level, the extant literature mainly focuses on exploring the role of ULCG policies in pollution emissions [4]. The promulgation of a ULCG policy is an important manifestation of attention on ULCG by government organisations. Low-carbon pilot policies help improve local air quality [5], reduce local carbon emissions [6], and increase carbon emissions from neighbouring provinces [7]. Technically, with the development of emerging technologies, such as big data and blockchain, technological progress has become the primary driving force behind declines in carbon intensity [8]. To some extent, digital construction promotes the disclosure of environmental information, which improves the efficiency of urban green emissions reductions [9]. At an environmental level, the institutional environment influences and dominates interactions between internal factors and the operational mode of the entire system of ULCG decisions by the government [10].

Although the studies mentioned give a certain degree of importance to ULCG, they are all conducted from a single perspective, such as government policy, technological innovation or the institutional environment [11,12]. ULCG is a dynamic process. The independent role of different elements along various dimensions limits managers' understanding of the synergies and matching effects of multiple factors [13].

Based on a configurational perspective [14], this paper constructs a theoretical ULCG model along the three TOE dimensions by applying the TOE framework, which, while being flexible, practical, operational, and adaptable to the complexities of the research problem, is a comprehensive analytical framework based on innovation diffusion theory and the technology authorisation model [15]. Particularly, combining the TOE framework and ULCG allows for analysis of the antecedents that drive improvements in ULCG along the three dimensions of the TOE framework. This makes it more conducive to exploring multiple paths that match the complex environment of ULCG when dealing with the internal causal complexity of governance models affected by various environmental factors.

Fuzzy-set qualitative comparative analysis (fsQCA) is used to conduct a multipath analysis of ULCG, revealing various strategies and configurations to address the complex challenges of ULCG [16]. The following issues are mainly addressed in this study: Firstly, a necessity analysis is conducted on the six preset antecedent conditions to observe whether there are necessary conditions that affect the low-carbon governance of high/low-level cities and the degree of necessity of these conditions. The study aims to resolve the issue of whether there exist bottleneck conditions that affect the low-carbon governance of high/low-level cities. Secondly, the sufficiency of the antecedent configuration is analysed using the fuzzy set qualitative comparative analysis (fsQCA). Various configurations of low-carbon governance in cities are deeply analysed to effectively distinguish the core and marginal conditions that affect the progress of low-carbon governance. This is to clarify the complex mechanisms of different conditions that influence low-carbon governance in cities and to resolve questions such as what paths are required to achieve high-level low-carbon governance in cities and what are the combination mechanisms of these paths. Finally, the study conducts a regional path analysis of the eastern, central, and western regions of China to explore the combinations of factors within different regional paths, as well as the differences between the regions. Based on the differences in China's resource endowments and economic development levels, the study aims to resolve the question of what differences exist in the low-carbon governance paths of the eastern, central, and western regions of China in the regional study.

This study makes several contributions. First, it applies the TOE framework to solve the ULCG problem based on the configuration of this framework. Further, it discusses the influencing factors and joint effects of ULCG for the three aspects of TOE, making up for the lack of a ULCG research perspective [17]. Second, from a configuration perspective, this study applies the fsQCA method to discuss various ULCG governance modes and analyse the internal relationship of each. This study extends traditional research on the univariate–ULCG relationship by applying measurement methods to comprehensively research the relationship between multivariate effects and ULCG from a configuration perspective. To an extent, it solves the key problem of achieving high-level ULCG within a complex environment. Finally, it reveals antecedent configurations of high-level ULCG paths within various regions, contributing to a deeper understanding of the paths and complex mechanisms of ULCG. In the diversified governance model, the government provides an important theoretical basis and practical guidance for ULCG. Managers can develop targeted policies and measures based on sub-regional path research findings.

## 2. Literature Review and Theoretical Framework

### 2.1. Literature Review

Technical means of ULCG. At the environmental level, some scholars have observed that low-carbon technology is essential for promoting carbon emission reduction [18,19]. Biomass fuel, for example, can effectively reduce the carbon intensity of the transport

sector [20]. Through digital reform and data supervision, the government can effectively promote modern ULCG [21]. Therefore, the Chinese government focuses on developing low-carbon information platforms and is dedicated to promoting the transformation of ULCG through systematic thinking [9]. However, the gradual application of modern information technology in ULCG requires not only the technology governance system and governance capacity at the macro level but also digital governance transformation and technological facility innovation at the micro level [22]. Wang and Sun [23] revealed that the incomplete and unclear accounting results of greenhouse gas emissions faced by low-carbon cities could be solved through digital technologies such as data analysis and mining or building an information infrastructure platform to guide the process of ULCG. According to Balsari et al. [21], digital technology, as the main software facility for government governance, can provide a flexible and efficient service platform for the government and effectively enhance its competitiveness. However, the current digital regulatory and information disclosure policies are imperfect, and there are defects in emphasising enterprise information disclosure and ignoring government information disclosur [24,25]. Therefore, driven by the concept of win–win cooperation, a top-level design of 'technology + management' has been formed to improve the integrity, synergy, and sustainability of digital technique [26].

Organisational Policies for ULCG. At an organisational level, most scholars attribute the factors influencing ULCG results to environmental policies. Environmental policies substantially represent government attention to ULCG. For example, Zhou et al. [3] stated that the low-carbon city pilot policy is crucial for China to tackle climate change and realise low-carbon transformation. Through empirical tests, several scholars have concluded that China's low-carbon pilot policies have significantly promoted regional growth [27] and a green economy [28]. According to resource allocation theory, when the government pays great attention to an event, it receives increased policy and financial support. These environmental policies all reflect governmental attention to ULCG. However, the causal relationship between a current environmental policy and UCLG is uncertain. It mainly encompasses two views, policy promotion and suppression. On the one hand, it is believed that China's pilot cities can effectively reduce carbon emissions comparatively more than non-pilot cities [6]. Specifically, implementing low-carbon policies can improve urban air quality and the level of regional green development, thus significantly reducing urban carbon intensity [5]. On the other hand, findings indicate carbon emission problems from the low-carbon pilot policy. Although policy implementation reduces the carbon emissions intensity of pilot cities, the increased carbon emissions of non-pilot cities increase overall carbon emissions [29]. Beyond environmental policies, fiscal policy is indispensable in public policy for coping with climate change and controlling environmental pollution. It plays a supporting, leading, promoting, and safeguarding role in constructing an ecological civilisation [30]. Jiang and Deng [31] used provincial data from China as the research sample, finding through empirical analysis that increased non-economic public expenditures can reduce regional carbon emissions. Tian [32] proposed that promoting synergy between fiscal and low-carbon policies could better promote ULCG.

External environment of ULCG. At the environmental level, most existing studies focus on how urbanisation affects ULCG. Although urbanised development is a significant factor influencing the urban low-carbon development environment, its complexity has led to a differentiated research conclusion. These research results show that the relationship between urbanisation level and UCLG is mainly positive [4] or negative [33]. For example, Yuan and Sun [34] empirically found current urbanisation levels for provinces with higher income levels to have produced significant emission reduction effects. In contrast, underdeveloped regions are still in the stage of accelerated carbon emissions in the urbanisation process. Hu [35] found that urbanisation has an inhibitory effect on carbon emissions in the Yangtze River Delta urban agglomeration and a promotional effect on the Beijing and Tianjin urban agglomerations in Hebei Province. Moreover, the scope of this role has significantly increased with the continuous progress of urbanisation. Although ur-

banised development is a significant factor influencing the urban low-carbon development environment, its complexity has led to a differentiated research conclusion.

In summary, the literature mentioned found two deficiencies. First, the ULCG level is related to governmental concerns at the organisational level and is affected by current data technology, technical infrastructure, and urbanised development at the environmental level. However, the literature mainly discusses the causal symmetry of a single variable. In China's practice scenario, linkage effects often occur among multiple factors, resulting in single initiatives affecting the entire body. Therefore, it is important to further combine the synergy of TOE elements to explore the complex theoretical mechanism and dynamic combination path of ULCG. Second, from an environmental standpoint, the literature mentioned found no involvement by factors of the level of marketisation. However, in recent years, the level of marketisation in China has risen steadily, and a carbon trading market has come to be established. Therefore, from the perspective of the marketisation level, combined with the role of technical and organisational factors, we can explore the comprehensive impact of ULCG more comprehensively. From a configuration perspective, this paper combines the three TOE factors, so the government can more clearly understand the connotations of ULCG and select differentiated governance paths based on regional characteristics to implement the low-carbon concept into urban planning, construction and all aspects of management.

### 2.2. TOE Framework

The TOE framework is a comprehensive analysis approach based on technology application scenarios [15] that can significantly improve the efficiency of urban emission reduction by utilising information technology. Initially, it was used to analyse the factors influencing enterprise innovation technology [36]; however, with the continuous evolution of the framework, it has been widely used in organisational management [37], e-governance [38], green technology, and urban public governance [39]. The framework has good scalability in variable selection because of its flexibility, practicality, and operability. Researchers can refine it based on various research objects and fields [38]. Recently, the TOE framework has also solved complex governance problems, such as China's regional [39] and urban problems [40]. The research framework of these problems can be adjusted according to the complexity of urban problems. Therefore, based on the TOE framework, this study can effectively build a configuration model of ULCG from the three TOE dimensions.

The TOE framework particularly focuses on the causes and influencing factors of basic organisations' innovation from the three perspectives of TOE [15]. The technical influencing factors refer primarily to the characteristics of the technology itself, such as the use status of existing technology and the characteristics of adoptable technology [41]. Organisational factors are primarily concerned with the internal attributes of the organisation, that is, whether the organisation can choose the appropriate innovation strategy based on its characteristics, such as organisational resources, scale and structure, technical management ability and financial resources [41]. In the interaction between technology and organisation, the primary focus is on whether the technology and organisational structure are compatible, coordinate application capability between organisations, and increase potential organisational benefits [42]. The proposal of environmental conditions, which includes an organisation's market structure, resource endowment or pressure or motivation, is a relatively new concept when compared with other theories from the same period [42].

### 2.3. Theoretical Model

Technical conditions. This study establishes two secondary conditions in the technical dimension: the level of development of big data and technical infrastructure. In recent years, the Internet has entered version 3.0 of its development [43], and digital technology has become an essential tool for reducing carbon emissions. Modern ULCG can be effectively advanced through digital reform and data regulation, among other means [21]. Consequently, the Chinese government prioritises the importance of a low-carbon infor-

mation platform and is committed to promoting the informatisation transformation of ULCG through the application of systematic thought [9]. However, the gradual application of modern information technology to ULCG must be supported by an information technology platform and big data development technology [22]. Coase's transaction cost theory highlights that reducing transaction costs is conducive to generating government policies or systems and can better coordinate organisational behaviour [44]. In the era of big data, constructing technological infrastructure can effectively reduce the transaction and information costs of government low-carbon decision-making, solve the problem of information asymmetry in the process of ULCG through information feedback, reduce the risk of policy implementation failure and provide a successful experience for decision-making [45]. Simultaneously, the continuous development of big data technology provides important technical means for low-carbon behaviour monitoring and information accounting and evaluation. Innovative use of digital technology can improve the accuracy and effectiveness of market-oriented means, such as carbon tax, trading, and subsidies; therefore, reducing the investment, financing, and social and economic costs required for green and low-carbon transformation so as to maximise the benefits of ULCG.

Organisational conditions. This study establishes two secondary conditions along the organisational dimension: attention distribution and financial resource support. Bounded rational decision-making believes that attention is a scarce resource and that the attention distribution of decision-makers determines their corresponding behaviour choices [45]. Different attention distribution will lead to different decisions. Therefore, attention distribution plays a crucial role in government agenda-setting and policy implementation [46]. Typically, central or higher levels of government can base hiring decisions on the performance of local officials [47]. Therefore, when a higher government promotes an innovation policy, lower officials are influenced by promotion incentives and pay greater attention to the policy's implementation [38]. The construction of a low-carbon city is a key project based on the dual-carbon objective and a 'top' project highlighting achievements in government governance. Based on the available financial resources, the municipal government can diligently implement the provincial or central government's decisions and arrangements and formulate pertinent policies according to the city's actual situation. In practising environmental governance, the government's attention will constantly change with the regional green development process. For example, before the carbon peak in 2030, the government will focus on 'how to achieve the carbon peak' and formulate relevant policies to achieve this goal. After the carbon peak in 2030, the government's attention will shift to 'how to achieve carbon neutrality'. Research indicates that implementing low-carbon policies is conducive to improving local air quality, thereby significantly lowering the urban air pollution index [5]. Moreover, it can significantly promote improvements in the regional industrial structure [48] and urban green total factor productivity [49,50]. Finance is the foundation and an important pillar of urban governance. Financial support, tax system, and government procurement are all important policy tools to promote carbon peaking and neutrality. Financial resources support can promote technological innovation, adjust the structure of financial expenditure and improve the city's total factor energy efficiency towards achieving the goal of reducing carbon emissions [50]. According to resource allocation theory, resources always show relative scarcity, which requires the rational allocation of limited and scarce resources. When the government's financial resource capacity is weak, resources are used for necessary public expenditures to meet the most basic and critical public demands. Financial supply is accordingly reduced for the technical public service of urban low-carbon construction.

Environmental conditions. In the environmental dimension, this study specifically establishes two secondary conditions: marketisation and urbanisation levels. The marketisation level can effectively reflect the current market structure and transaction level. Its purpose is to reorganise and change the market environment to make the operation more effective [47]. According to transaction cost theory, the improvement of the market-oriented level can effectively enhance the institutional environment, the intensity of information

disclosure, and the construction of an information transparency mechanism, thereby reducing the transaction cost of low-carbon information and improving the accuracy and convenience of low-carbon information collection and processing [36]. The rapid progression of urbanisation has had a profound impact on carbon intensity, which has become the focus of research for a large number of domestic and international scholars. At the initial stage of urbanisation, the serious negative externalities owing to population agglomeration and extensive economic development have led to severe urban resource shortage and environmental pollution, which is reflected in the positive impact of carbon intensity by the level of urbanization [51,52]. Undoubtedly, this pressurises ULCG. However, with the advancement of urbanisation, the agglomeration effect has expanded the positive externalities of human capital, especially the agglomeration of high-level talents, which can continuously accumulate the urban capital stock at the macro level and improve the regional low-carbon research and development (R&D) efficiency and urban low-carbon management efficiency. Subsequently, it promotes the ULCG level of the whole region, which is reflected in the urbanisation level's reverse impact on carbon intensity [33].

Interaction from a configuration perspective. The establishment of interactions between conditions is mainly applied through transaction cost theory, which asserts that reducing transaction costs is conducive to developing government policies or systems and improving the coordination of organisational behaviour [44]. For example, in cities with relatively high levels of big data development, digital technology can effectively solve the government's information asymmetry problem, reduce transaction costs in low-carbon information collection and processing and improve policymaking and organisational operation efficiencies, thus accelerating ULCG policy formulation. In cities with relatively high marketisation levels, establishing and improving the carbon trading market promotes the construction of an online trading platform, reducing the transaction costs of big data technology development and affecting the operational mode of the entire ULCG system. In cities with relatively high urbanisation levels, the agglomeration effect of many talents has significantly improved technological innovation levels within cities. In addition, organisational carbon policies have improved the intensity of environmental regulation by local governments, motivated enterprises to increase their R&D investments, and thus improved the innovation levels of low-carbon technology. The combination of TOE factors complexly described the result of high-level ULCG. The interaction of factors at three levels compensates for the deficiency of using a single aspect in ULCG, which affects the city-level low-carbon transformation process and governance decisions [53]. Figure 1 illustrates the specific configuration model.

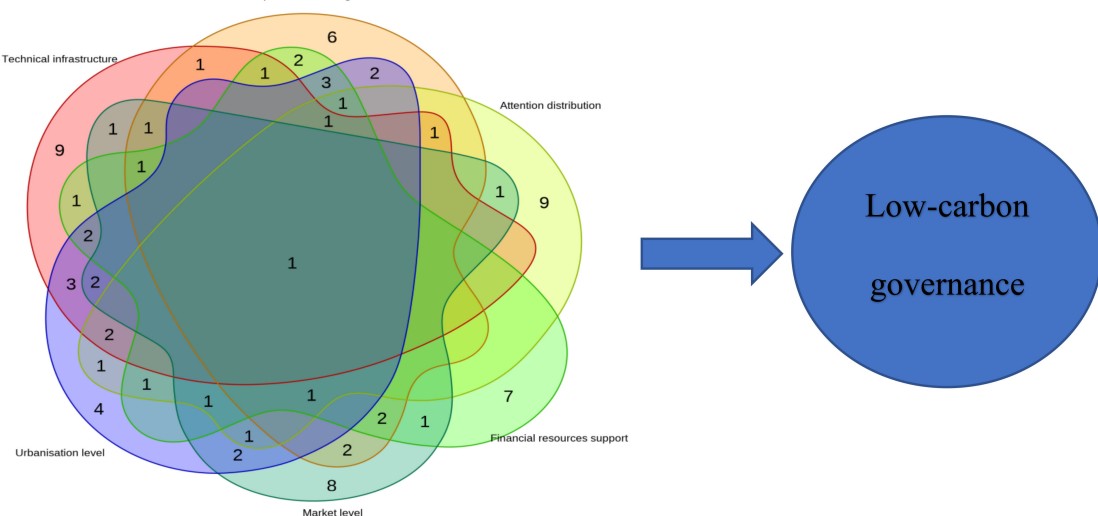

**Figure 1.** Configuration model. Note: The numbers in the figure represent the number of corresponding cases (cities).

## 3. Methods

Using the fsQCA method, this study first analysed the necessity of six prerequisites for ULCG to determine whether high-level/low-level ULCG met the necessary conditions. Second, it analysed the sufficiency of these six antecedents, explored different combination modes of ULCG paths leading to high-level cities and determined strategies needed to achieve high-level ULCG under complex operating conditions.

### 3.1. Qualitative and Comparative Analysis

Qualitative and comparative analysis (QCA) was first developed by Ragin (1987) [17]. This method is based on holism, which considers that the research case comprises multiple antecedents and focuses on resolving the complex causal relationship between the condition configuration and the result [42]. The essence of the QCA method is 'result-driven' and applies the 'configuration effect', which was initially mainly applied to politics, sociology and management [46,54]. It has expanded recently to include organisational and strategic management and technology applications [42,54]. It is mainly used to solve the complex cause-like relationship problem [42]. This paper introduced the fsQCA method for several reasons. First, ULCG antecedents are interdependent rather than independent, necessitating a holistic and combined explanation for the high level of ULCG. Compared with the traditional regression analysis that discusses the net effect of a single variable, the fsQCA method can better explain the 'configuration problem' among many factors of urban governance, reflect the dynamic complementarity between antecedents and effectively identify the interaction between various factors. Second, the fsQCA method combines the benefits of qualitative and quantitative analyses. It applies to 10/15–50 small- and medium-sized sample cases, as well as large sample case studies with over 100 cases. It is consistent with the small- and medium-sized sample case studies carried out in 35 key cities in this paper. This study combines qualitative and quantitative analyses to examine the condition configurations of intra- and inter-group relationships and find the 'complexity' issue and the 'different approaches lead to the same goal' phenomenon among multiple elements. Third, when compared with configuration research methods, such as cluster analysis and factor analysis, QCA's most significant advantage is that it can effectively identify the dynamic complementarity, configuration equivalence and causal asymmetry between the leading cause and factor conditions [40]. Finally, when compared with other QCA methods (csQCA and mvQCA), fsQCA can fully capture the subtle changes of antecedent conditions at different levels [55]. Considering the complex mechanism of the research object in this study and as the causal conditions were mostly continuous variables, the fsQCA method was used to explore the ULCG level of the city.

### 3.2. Sample Selection and Data Source

This paper's case sample comprised 35 key cities, including provincial and sub-provincial cities, municipalities directly under the central government and cities separately listed in the state plan (excluded due to missing data in Lhasa). The selection criteria for sample cities were mainly based on overall homogeneity and maximum internal heterogeneity (According to the documents of the National Development and Reform Commission and the National People's Congress, divide cities in eastern, central, and western China (corresponding to each province). East China: Shenyang, Beijing, Nanjing, Tianjin, Shanghai, Ningbo, Hangzhou, Fuzhou, Guangzhou, Shenzhen, Shijiazhuang, Xiamen, Jinan, Qingdao, Haikou, and Dalian. Central China: Hefei, Nanchang, Wuhan, Zhengzhou, Changchun, Harbin, Taiyuan, and Changsha. Western China: Chengdu, Guiyang, Kunming, Nanning, Xi'an, Chongqing, Xining, Yinchuan, Lanzhou, Urumqi, and Hohhot.) [55]. From the perspective of sample homogeneity, we selected cities above the sub-provincial level with good regional representation and that belonged to the scope of low-carbon city construction. In terms of data selection, we specifically selected data from 2019 for measurement. We considered the China Urban Statistical Yearbook 2020 the primary data source to calculate the year-end GDP and total population of key cities, widely used in urban governance

research. Simultaneously, data from the China Statistical Yearbook and *China* Energy Statistical Yearbook were used to determine urbanisation levels and energy consumption. The big data development level measurement was obtained from the commercial, civil and government use indexes in the China Big Data Development Report; the market-level data were based on the market-level index of municipalities and key cities.

### 3.3. Outcome Measurement

Low-carbon governance level. Urban low-carbon development reduces carbon dioxide emissions and improves carbon solidification and neutralisation capacity. Based on the practise of Wu and Guo [7], we used urban carbon intensity (the ratio of total urban carbon dioxide emissions to actual regional GDP) to measure the ULCG level of 35 cities in 2019. The lower the urban carbon intensity, the better the ULCG effect. Simultaneously, considering available city-level data, total urban carbon dioxide emission measurement was evaluated primarily by natural gas (manufactured and natural gas) consumption, LPG consumption and carbon emissions generated during power generation. We used data from the China Urban Statistical Yearbook 2020 and the China Energy Statistical Yearbook 2020.

### 3.4. Conditional Measurement

In the technical dimension, this study sets up two secondary conditions: the development level of big data and the technical infrastructure. For the technical infrastructure, according to Tan et al. (2019) [42], the average number of Internet broadband access ports (10,000) per million populations in 35 key cities in 2019 was used as the measurement index. Data were obtained from the China Urban Statistical Yearbook 2020. Meanwhile, regarding the development level of big data, following Tao et al. [36], the research on the development level of big data was assessed using the big data development comprehensive index data of 31 key cities and four municipalities directly under the central government in 2019 published in the 2020 Big Data Blue Book: China Big Data Development Report No. 4.

In terms of the organisational dimension, this study establishes two specific secondary conditions: attention distribution and financial resources support. Attention distribution: By referring to the measurement method of Tan et al. [42], we use the time interval between the response policy formulated by the local government and the policy issued by the central government to calculate. In particular, the official document 'preparation' of the zoning control policy of the ecological protection red line, environmental quality bottom line, resource utilisation online and ecological environment access list (three lines and one order) issued by the central government on 25 December 2017 was considered the starting time. It was found in the data collection that each city responded after the corresponding provincial government issued and formulated the document. Moreover, the time for the provincial government to issue the preparation document was inconsistent. Therefore, to ensure the results' accuracy, we measured each province's response time interval to the policy. Financial resource support: Using the measurement method of Zhao et al. [39] for reference, the public budget expenditure per million populations of 35 key cities in 2019 was used as the measurement index to measure government financial expenditure. We used data from the China Statistical Yearbook 2020.

This study establishes marketisation and urbanisation levels as specific secondary conditions in the environmental dimension. Marketisation level: The level and degree of regional market-oriented development are represented by the marketization index. The marketization index consists of five aspects, each reflecting a different aspect of marketisation: 1. The relationship between the government and the market. 2. The development of a non-state-owned economy. 3. The degree of development in product markets. 4. The level of development in factor markets. 5. The development of market intermediaries and legal environment. The measurement of the level of marketisation is the weighted average of the above five indicators. This measurement method draws on Fan Gang's "Marketisation Index Report" [56]. Urbanisation level: Based on the measurement method of Dong and Li [4], we measured the urbanisation level using the ratio of the urban population to the

total population at the end of 2019. We used data from the China Statistical Yearbook 2020. The descriptive statistical results of the variables are shown in Table 1.

**Table 1.** Descriptive statistics.

| Variable | Mean | SD | Min | Max |
|---|---|---|---|---|
| Technical infrastructure construction | 54.761 | 17.707 | 25.543 | 97.290 |
| Development level of big data | 43.250 | 17.884 | 6.740 | 75.960 |
| Attention distribution | 1069.114 | 106.418 | 871.000 | 1369.000 |
| Financial resources support | 221.698 | 147.178 | 99.900 | 826.700 |
| Marketisation level | 7.812 | 2.170 | 3.610 | 11.400 |
| Urbanisation level | 0.626 | 0.098 | 0.485 | 0.883 |
| Urban low-carbon level | 337.2324 | 303.900 | 133.970 | 1717.800 |

*3.5. Calibration*

Case data must be calibrated before configuration analysis [57]. In the fsQCA method, each condition and result are considered as a set, and each case has a membership score in these sets. Giving a set membership score for a case is calibration [58]. This paper adopted the direct calibration method (using the objective quantile value to determine the anchor point), consistent with prior research [59,60]. there are three specific anchor points: 'full membership' (very consistent), 'intersection' (agree and disagree), and 'full non-membership (very inconsistent) [61]. First, when calibrating the conditions, we used the substantive knowledge of the prevalidated scale anchor points to set the full membership and non-membership of the other five conditions except for attention allocation to 95% and 5% quantiles, respectively (the variable distribution is closely aligned with the anchor points) [60,62]. Considering that attention distribution is measured by the time interval between local governments' responses to central policies, the shorter the time interval is, the stronger the government's attention is. Therefore, reverse calibration was adopted for attention distribution, and the complete subordination and non-subordination were set to 5% and 95% quantiles, respectively [39]. Second, the lower the urban carbon intensity, the greater the ULCG effect, with the scale anchor points of the data having different distributions [60]. Considering these points, the second calibration method was adopted to set the ULCG level's full membership and non-membership to 10% and 90% quantiles, respectively [60,63], when calibrating the outcome variable (ULCG). Finally, we set the sample data mean as the intersection calibration standard for all conditions and results [63,64]. Additionally, to ensure the accuracy of the data, three decimal places were reserved in the calibration process. The specific calibration data for each condition and result are presented in Table 2.

**Table 2.** Fuzzy-set calibrations.

| Category | Conditions and Results | Calibration | | |
|---|---|---|---|---|
| | | Fully Out | Crossover | Fully In |
| Result variable | Urban low-carbon level | 155.719 | 337.232 | 564.001 |
| Technical conditions | Technical infrastructure | 92.306 | 54.761 | 31.533 |
| | Development level of big data | 74.889 | 43.250 | 20.766 |
| Organisation conditions | Attention distribution | 1 207.000 | 1 069.110 | 886.400 |
| | Financial resources support | 538.102 | 221.698 | 111.300 |
| Environmental conditions | Marketisation level | 10.960 | 7.812 | 4.556 |
| | Urbanisation level | 0.844 | 0.626 | 0.490 |

## 4. Results

*4.1. Necessity Condition Analysis*

Necessary and sufficient causality are two emerging methods of causal interpretation, where essential condition causality refers to a condition that always exists in the

outcome [65]. If the necessary conditions are included in the true table analysis, it may be simplified in the parsimony solution incorporating the 'logical remainder' [59]. Therefore, the necessary condition analysis is required before the configuration analysis [66]. The standard of the necessary conditions test determines the level of consistency, and this paper is based on the previous research standards [60]. The consistency-level threshold was set at 0.9, and the 'necessity' test of whether the conditions constitute the ULCG performance level was conducted through fsQCA 3.0 software [14].

Specific analysis results are presented in Table 3. Six conditions of high-level ULCG performance and ULCG performance consistency were less than 0.9, so they cannot constitute a necessary condition. It reflects the complexity of ULCG, further illustrating the need to focus on the technological, organisational and environmental conditions of the results' synergistic effect.

**Table 3.** Analysis of necessary conditions for urban low-carbon governance in the fsQCA.

| Conditions | High Level of Urban Low-Carbon Governance | | Low Level of Urban Low-Carbon Governance | |
| --- | --- | --- | --- | --- |
| | Consistency | Coverage | Consistency | Coverage |
| Technical infrastructure construction | 0.476 | 0.743 | 0.670 | 0.495 |
| ~Technical infrastructure construction | 0.676 | 0.812 | 0.652 | 0.371 |
| Development level of big data | 0.624 | 0.920 | 0.441 | 0.308 |
| ~Development level of big data | 0.531 | 0.667 | 0.885 | 0.527 |
| Attention distribution | 0.664 | 0.829 | 0.601 | 0.355 |
| ~Attention distribution | 0.484 | 0.719 | 0.747 | 0.500 |
| Financial resources support | 0.394 | 0.814 | 0.474 | 0.464 |
| ~Financial resources support | 0.741 | 0.748 | 0.811 | 0.388 |
| Marketisation level | 0.642 | 0.784 | 0.460 | 0.296 |
| ~Marketisation level | 0.484 | 0.654 | 0.805 | 0.516 |
| Urbanisation level | 0.526 | 0.874 | 0.523 | 0.412 |
| ~Urbanisation level | 0.620 | 0.750 | 0.747 | 0.427 |

Note: ~ means the absence of. For example: ~ Attention distribution = absence of Attention distribution.

### 4.2. Sufficiency Analysis Performance

Sufficient conditions refer to the combination of antecedent conditions that fully produce results [65]. According to the extant literature, the consistency level of adequacy analysis should not be lower than 0.75 [58]. Therefore, scholars adopted different consistency thresholds according to different research situations, such as 0.76 [55] and 0.80 [36]. The setting of the frequency threshold must be adjusted according to the sample size. The frequency threshold of small and medium samples can be 1, while the frequency threshold of large samples should be greater than 1 [58]. Based on the existing research results and the distribution of case data, we set the consistency threshold, frequency threshold and PRI to 0.8, 1 and 0.7, respectively.

From the five configuration results presented in Table 4, the consistency levels of the single and overall solutions were higher than 0.8 and higher than the acceptable minimum standard of 0.75 [58]. The overall solution consistency and coverage are 0.930 and 0.579, respectively. Therefore, the six configurations can be regarded as sufficient conditions to reduce the urban carbon intensity. The following three modes exist.

The core existence conditions of configurations S1a, S1b and S1c of configuration S1 are the development and market levels of big data. The three group configurations of S1 show that under the core role of the joint improvement of big data and marketisation, it can effectively reduce the urban carbon intensity and improve the ULCG level. Therefore, the three groups of configurations were classified as the ULCG mode dominated by 'big data + market'. In configurations S1a, S1b, and S1c, the levels of consistency were 0.915, 0.950 and 0.993, the raw coverage ratios were 0.339, 0.248 and 0.292, and the unique coverages were 0.052, 0.042, and 0.006, respectively. The original coverage of configuration

S1a is the highest among the five configurations, indicating that this path covers the largest number of urban samples. Therefore, we should concentrate on how big data development and marketisation levels jointly play a central role, with the assistance of financial resources and the level of urbanisation. Additionally, we find that the attention distribution conditions in configuration S1a are blank, indicating that they may exist. Therefore, it is impossible to determine the full extent of attention allocation condition effects. Under the S1 mode, on the one hand, by encouraging an improved marketisation level, the transaction cost of big data technology development can be reduced. The acquisition speed of government low-carbon data can then be improved. On the other hand, big data technology development can promote the establishment of a carbon trading market and low-carbon information platform, thus improving the marketisation level. The interaction of both conditions as core factors can effectively promote the process of ULCG. The representative cities included Shanghai, Hangzhou, Nanjing, etc., as shown in Figure 2. These cities are mainly concentrated in the Yangtze River Delta Economic Zone, with a high level of economic development and openness. Take Shanghai, for example. Shanghai has hosted the China International Import Expo for four consecutive years since 2018. During this time, it established a special zone for low-carbon energy, technology and environmental protection to create an intelligent low-carbon platform using high-performance computing, high-frequency links and advanced display technology. Meanwhile, in July 2021, Shanghai established a low-carbon innovation research centre to promote the integrated development of industry, university and research, as well as the deep integration of capital and low-carbon disciplines based on the market.

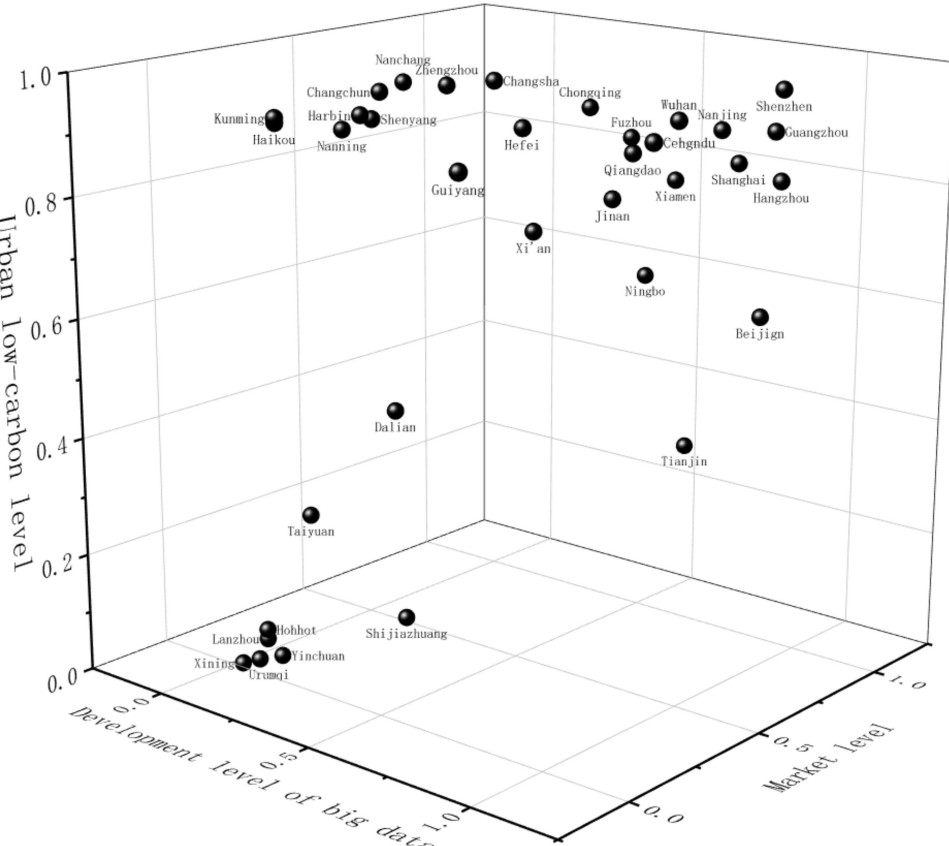

**Figure 2.** Explanation case of condition configuration 1.

**Table 4.** Configurations for achieving high-level urban low-carbon governance (fsQCA).

| Configuration | S1: The Low-Carbon Governance Model Dominated by 'Big Data + Market' | | | S2: The Low-Carbon Governance Model Dominated by Big Data | S3:The Low-Carbon Governance Model Dominated by Market |
|---|---|---|---|---|---|
| | S1a | S1b | S1c | S2 | S1a |
| Technical infrastructure | ⊗ | ⊗ | ● | ⊗ | ⊗ |
| Development level of big data | ● | ● | ● | ● | ⊗ |
| Attention distribution | | ⊗ | ● | • | • |
| Financial resources support | • | ⊗ | • | ⊗ | ⊗ |
| Marketisation level | ● | ● | ● | ⊗ | ● |
| Urbanisation level | • | | | ⊗ | ⊗ |
| Consistency | 0.915 | 0.950 | 0.993 | 0.952 | 0.953 |
| Raw coverage | 0.339 | 0.248 | 0.292 | 0.249 | 0.245 |
| Unique coverage | 0.052 | 0.042 | 0.006 | 0.053 | 0.047 |
| Overall solution consistency | | | | 0.930 | |
| Overall solution coverage | | | | 0.579 | |

Note: ●= core causal condition present; • = peripheral condition present and ⊗ = peripheral condition absent.

In configuration S2, the big data development level is the core existence condition, attention distribution is the peripheral existence condition, and other conditions are peripheral absent conditions. This shows that improvement in the big data development level, combined with increased government attention, can effectively reduce the urban carbon intensity compared with other conditions. Accordingly, configuration 2 was classified as the ULCG model dominated by big data. The levels of consistency, raw coverage and unique coverage were 0.952, 0.249, and 0.053, respectively. Under the S2 mode, cities can focus on developing local big data technology. With the assistance of attention distribution, combined with the core role of big data technology, it can effectively promote the low-carbon process of cities. In cities where big data technology is high, the cost of obtaining relevant information on ULCG can be effectively reduced by fully exploiting technical advantages. Driven by the core of big data technology, we will try our best to raise the government's attention to ULCG and achieve a high level of ULCG. In contrast, if the government wants to conduct ULCG through this model in cities where big data technology is low, it must do its best to focus on digital technology and invest more resources in it. The representative cities included Guiyang, Hefei, Chengdu, and Xi'an, as shown in Figure 3. Using Chengdu as an example, the 2017 Chengdu low-carbon city pilot implementation plan emphasises the importance of focusing on the development of emission source management, low-carbon assessment and analysis and low-carbon target assessment systems, as well as building the city's low-carbon development management platform and strengthening the informatisation construction of data monitoring, management, development, and service in the carbon emission-related field. According to the 2018 Chengdu Internet development report, Chengdu has achieved rapid growth and continuous innovation in big data, artificial intelligence and new technology development, effectively improving the government's ability to obtain ULCG information and promoting the development of a low-carbon science and technology public service system.

In configuration S3, the marketisation level is the core existence condition, attention distribution is the peripheral existence condition, and other conditions are peripheral absent conditions. The consistency, raw coverage, and unique coverage levels were 0.953, 0.245, and 0.047, respectively. While the marketisation level effectively improves, combined with increased government attention, it will significantly reduce the urban carbon intensity and improve ULCG performance. Thus, configuration S3 was classified as the ULCG model dominated by the market. Under the S3 mode, the level of marketisation plays an important role. In cities with high levels of marketisation, fully exploiting this environmental advantage can effectively reduce the cost of big data development and digital infrastructure construction at the technical level and promote the establishment of a carbon trading market. Under the core impetus of the marketisation level, the government should attempt to increase attention on ULCG to achieve a high ULCG level. In contrast, if

we want to conduct ULCG through this mode in cities with low levels of marketisation, we must focus on improving the local institutional environment and the level of marketisation. The representative city included Changsha. Changsha is the provincial capital of Hunan Province. It is an important central city in the middle reaches of the Yangtze River, and its marketisation level is relatively high. The carbon exchange of Changsha was established in 2015 and is one of the earliest seven carbon exchanges in China. In 2017, Changsha became the third batch of low-carbon pilot cities issued by the National Development and Reform Commission. The Changsha Municipal Development and Reform Commission proposed the Low-Carbon Development Plan (2018–2025) (draft) to reach the carbon peak five years ahead of China's target, indicating the significance of the government.

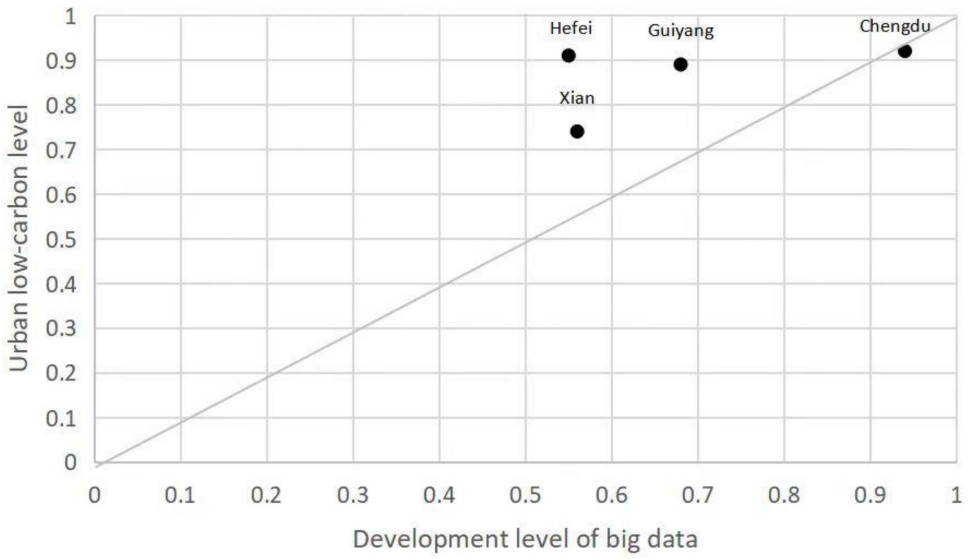

**Figure 3.** Explanation case of condition configuration 2.

Configuration S2 is interchangeable with configuration S3. That is to say, when other conditions are the same, the big data development and marketisation levels in configuration S2 and S3, respectively, can be used as the core existence conditions to replace each other to achieve the effect of 'reaching the same goal by different routes'. When attention distribution is a marginal condition, cities lacking big data development can choose to increase their level of marketisation as a substitute, and vice versa. This alternative method of resource allocation enables rapid and efficient ULCG. Detailed substitution effects are illustrated in Figure 4.

Technical infrastructure construction+Attention distribution+Financial resources support+Urban low-carbon level

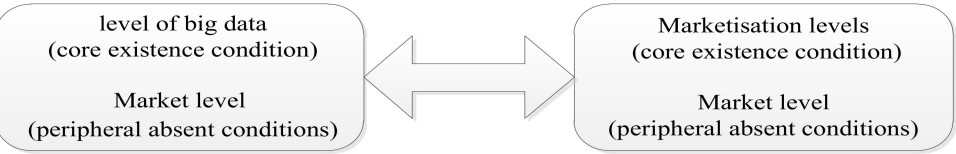

**Figure 4.** Alternative relationship between configurations 3 and 4.

In the five configuration groups, organisational conditions do not exist as core conditions. There are two primary reasons for speculation: First, the attention distribution conditions in configuration S1a are blank under the limit of the consistency threshold, indicating that they may exist and cannot be determined. Second, the two secondary organisational conditions individually or collectively appear in multiple configurations as marginal conditions, indicating that they play a particular role. Due to the small sample size, the organisational conditions do not reveal a core role; however, this does not imply

that there is no core role. This generally indicates that no core effect exists under the current sample size and consistency threshold, but it does not represent all configuration results.

### 4.3. Robustness Checks

Different consistency levels and calibration criteria can affect the number of logically minimised true table rows (configurations), affecting the results [60]. This article referred to the two criteria for judging the robustness proposed by Schneider and Wagemann [58], used the existing calibration methods and tested the results for robustness by changing the calibration interval and consistency threshold [39,42]. In particular, we replaced 5% and 95% with 6% and 96%, respectively, to adjust the calibration interval, and we adjusted the consistency threshold from 0.90 to 0.85. The robustness cheque results are shown in Tables 5 and 6. The results between configurations demonstrated weak changes, but they still had a clear subset relationship, indicating that the conclusions were robust.

**Table 5.** Configurations after changing the calibration interval (fsQCA).

| Configuration | 1 | 2 | 3 | 4 |
|---|---|---|---|---|
| Technical infrastructure | | | ⊗ | ⊗ |
| Development level of big data | ● | ● | ● | |
| Attention distribution | ⊗ | | • | • |
| Financial resources support | | • | ⊗ | ⊗ |
| Marketisation level | • | • | | ● |
| Urbanisation level | • | • | ⊗ | ⊗ |
| Consistency | 0.934 | 0.925 | 0.958 | 0.961 |
| Raw coverage | 0.306 | 0.389 | 0.288 | 0.310 |
| Unique coverage | 0.024 | 0.116 | 0.024 | 0.046 |
| Overall solution consistency | | | 0.930 | |
| Overall solution coverage | | | 0.579 | |

Note: ●= core causal condition present; • = peripheral condition present and ⊗ = peripheral condition absent.

**Table 6.** Configurations after changing the consistency threshold (fsQCA).

| Configuration | 1 | 2 | 3 | 4 | 5 | 6 |
|---|---|---|---|---|---|---|
| Technical infrastructure | ⊗ | ⊗ | • | ⊗ | • | ⊗ |
| Development level of big data | ● | ● | ● | ● | ● | ⊗ |
| Attention distribution | ⊗ | ⊗ | | | | |
| Financial resources support | ⊗ | | • | ⊗ | • | ⊗ |
| Marketisation level | ● | ● | ● | ⊗ | • | ● |
| Urbanisation level | | • | • | ⊗ | • | ⊗ |
| Consistency | 0.952 | 0.902 | 0.981 | 0.948 | 0.988 | 0.950 |
| Raw coverage | 0.261 | 0.256 | 0.221 | 0.268 | 0.301 | 0.232 |
| Unique coverage | 0.016 | 0.013 | 0.006 | 0.053 | 0.077 | 0.040 |
| Overall solution consistency | | | | 0.945 | | |
| Overall solution coverage | | | | 0.569 | | |

Note: ●= core causal condition present; • = peripheral condition present and ⊗ = peripheral condition absent.

### 4.4. Differentiation Path of ULCG in Eastern, Central and Western China

Due to the differences in economic level and resource endowment, there is clear regional heterogeneity in the performance level of ULCG. Currently, China faces the dual pressure of international carbon transfer and domestic carbon emission reduction, which is even more difficult for economically underdeveloped regions. Additionally, differences in institutional environments may also lead to varied effects of big data development, financial resource support and urbanisation on ULCG. Therefore, this paper divided the entire sample data into eastern, central and western categories and used the fsQCA method for differentiated path analysis. During the calibration, we set the consistency level to 0.97, 0.96, and 0.94; the frequency threshold was set to 1, and the PRI was set to 0.90, 0.85 and 0.90. Through the configuration analysis of the six conditions, we explored the ULCG paths in different regions using TOE, as detailed in Table 7.

**Table 7.** Configurations for achieving high-level urban low-carbon governance in the east, central, and western regions (fsQCA).

| Configuration | East China | | | | Central China | | | Western China |
|---|---|---|---|---|---|---|---|---|
| | **1** | **2** | **3** | **4** | **5** | **6** | **7** | **8** |
| Technical infrastructure | ● | ● | ⊗ | | ⊗ | ⊗ | ● | ⊗ |
| Development level of big data | • | ⊗ | ● | • | | | • | ⊗ |
| Attention distribution | | ⊗ | ⊗ | ● | ● | ● | ● | • |
| Financial resources support | • | ⊗ | ⊗ | • | ⊗ | ⊗ | • | ⊗ |
| Marketisation level | • | ⊗ | • | ● | ⊗ | | • | ⊗ |
| Urbanisation level | • | ⊗ | | • | ⊗ | ⊗ | ⊗ | ⊗ |
| Consistency | 1.000 | 0.989 | 0.935 | 0.970 | 0.947 | 0.947 | 1.000 | 0.972 |
| Raw coverage | 0.479 | 0.159 | 0.262 | 0.474 | 0.527 | 0.521 | 0.230 | 0.390 |
| Unique coverage | 0.054 | 0.048 | 0.097 | 0.038 | 0.013 | 0.040 | 0.031 | 0.390 |
| Overall solution consistency | | | 0.955 | | | 0.953 | | 0.390 |
| Overall solution coverage | | | 0.692 | | | 0.597 | | 0.972 |

Note: ●= core causal condition present; ⊗ = core causal condition absent; • = peripheral condition present and ⊗ = peripheral condition absent.

As can be seen from Table 7, there are clear differences in carbon emission reduction paths in eastern, central and western China. Overall, there were eight kinds of configuration; consistency was higher than 0.94, which is greater than the acceptable minimum standard of 0.75. The ULCG path in the eastern region comprises configurations 1a, 1b, 1c, and 1d. The core factors involved in TOE reflect the complexity and diversification of the antecedent configuration in the governance path of low-carbon cities in the eastern region. Because the eastern region's economy is developed, the urbanisation level is high. Simultaneously, the market-oriented and big data development levels are also relatively high, and the technical infrastructure is relatively perfect. Therefore, numerous types of paths exist in the eastern region, and the specific mode can be selected according to the actual situation of the region.

The ULCG path in the central region is configurations 2a, 2b, and 2c. The core existence condition of all three configuration groups is the attention distribution. The research revealed that attention distribution in the central region is central to ULCG. Compared with the eastern region, the economic development level of the central region is relatively low. There are numerous heavily polluting enterprises (such as the coal industry in Shanxi Province), and the geographical location is between the east and the west, which is prone to pollution spillover. However, the central region is rich in resources and has great market potential. Therefore, the state and local governments attach great importance to it, and local governments will do their best to promote ULCG.

The ULCG path in the western region is configuration 3. The big data development level is the core existence condition, attention distribution is the peripheral existence condition, and other conditions are peripheral absent conditions. It reflects the importance of the development level of big data for ULCG in the western region. Under the leading role of big data, assisting the allocation of government attention can effectively improve the ULCG performance level of the western region. However, most cities in the western region are economically underdeveloped and have relatively backward technology levels. Therefore, big data technology has a great space for development and can significantly improve the performance of ULCG. In summary, China's east, middle, and west regions are affected by technical level, organisational capacity, external environment and other factors resulting in significant differences in ULCG paths in different regions.

## 5. Discussion

Based on the TOE framework and using the fsQCA method to analyse 35 key cities in China as observation samples, this paper discusses the 'joint effects' of the six influencing factors of ULCG. Further, it reveals the core conditions and their complex interactive natures that affect ULCG.

## 5.1. Theoretical Implications

First, combining the TOE framework and ULCG expands the application of the TOE framework in ULCG and further refines research on ULCG conditions. The previous fields for applying the TOE framework involve research on regional issues and green innovation in China [39]. However, research on ULCG in detailed fields is scarce. This study correlates ULCG with the three dimensions of TOE, compensating for the deficiency of analysing ULCG from only a single perspective (e.g., finance or technology) [18]. Based on the TOE framework, combined with the present situation and characteristics of ULCG by the Chinese government, this study examines the linkage effects of TOE on ULCG. It provides a multifactor coupling path for improving the level of ULCG. Moreover, it expands the application of the TOE framework in ULCG, explores strategies closely aligned with ULCG, and comprehensively analyses the forms of ULCG. It enriches the width and breadth of the theoretical research on ULCG.

Second, from an allocation perspective, it enriches the causal complexity of ULCG research. This study discusses the differences between each configuration and reveals how to achieve ULCG under combinations of 'big data' and 'market-oriented' types. This means that under the leading role of various core factors, combined with assistance from other factors, cities can more easily achieve a high level of ULCG. For example, in the type of path dominated by big data, the development level of big data in cities has been improved, which can result in a higher level of ULCG with assistance from organisations and the environment. This discovery is a new attempt to explain the overall perspective of the path choice of ULCG. Compared with these previous empirical analysis methods, this study, from a 'configuration perspective', provides a more granular analysis of the mutual substitution effect and causal asymmetry between the six antecedents of ULCG [67].

Third, using fsQCA helps narrow the methodological gap in ULCG research. Hitherto—a common method to analyse the impact of various strategies and conditions on ULCG—is a traditional linear regression model based on the double difference [43,68] or a comprehensive control method [51,69]. This symmetrical approach oversimplifies complex environmental issues and cooperation among the various precursors of ULCG [70]. This paper uses the fsQCA method to discuss the necessary conditions for ULCG and the antecedent configuration of achieving high-level ULCG. However, previous studies have believed that single factors (such as technology level, urbanisation level, low-carbon pilot policies, etc.) significantly correlate with the effect of ULCG. However, this study found that these factors are unnecessary for high-level ULCG. For example, the lack of technology in S1a is a marginal lack of conditions. However, combining the core role of big data development and marketisation levels with the auxiliary role of financial resources and the urbanisation level can result in a significantly high ULCG level. The inspiration is that significant differences in TOE between cities do not prevent them from improving their levels of ULCG by combining various factors. This research compensates for the lack of research in the field of ULCG and helps narrow the methodological gap [60].

## 5.2. Practical Implications

Overall, the emphasis should be on linking and matching multiple TOE conditions to form a customised ULCG model.

First, according to the path analysis results, each city should adopt a governance mode according to the core conditions of the corresponding path. In mode S1, big data and marketisation are well coupled as the core causal condition present, promoting high-level ULCG in cities. Most case cities belong to this configuration, indicating that in the process of ULCG, the mutual benefit and symbiosis of technology and the environment should be emphasised. They must consider technological factors as the fundamental impetus for policy innovation and promote technological change in conjunction with innovation policies and the external environment. S2 is a low-carbon development path dominated by marketisation, demonstrating that under the leadership of the marketisation level and with the government's attention, it can effectively promote the ULCG process. Under this

model, local governments should maximise the government's ability to construct a service market, a carbon trading market, a multilevel trading platform and specific carbon sink policies to develop a low-carbon factor market. S3 is a low-carbon development path led by big data, which shows that under the guidance of the big data development level and combined with the government's attention, it can effectively promote the ULCG process of cities. Local governments should exert a low-carbon innovation, build low-carbon service platforms, build information distribution centres, reduce transaction costs of government ULCG and thus form cloud management centred on big data.

Second, according to the sub-regional study, the eastern, central and western regions should adopt differentiated governance modes according to local resource endowments and conditions. The eastern region should strengthen synergies among the three TOE elements and select a differentiated ULCG path and targeted governance model along with the regional resource endowment and economic development level. The central region should promote government attention to ULCG. The western region should try its best to make up for technical weaknesses, formulate special technical subsidies, create policy incentives for low-carbon development and promote the development process of big data. These measures can promote the balanced development of ULCG in the eastern, central, and western regions of China.

### 5.3. Limitations and Future Research

Although this paper discusses the path of ULCG from the perspective of configuration, which makes up for the net effect of a single factor from the contingency perspective, there are still shortcomings.

First, although qualitative analysis is supplemented in this paper, like other QCA research methods, it still faces the common challenge of deepening the qualitative analysis. Second, in terms of data collection, this paper only examines 35 representative key cities in China, which affects the universality of the conclusion to a certain extent. In the future, more sample data can be collected to conduct an in-depth analysis of the allocation path of ULCG. Moreover, only one-year static data is analysed, which limits the interpretation of the research conclusions in the time dimension. As the application of the QCA method in dynamic time change must be improved, the time-series QCA method can be reasonably developed to collect cross-time case data for research. Finally, in terms of data measurement, attention distribution is measured by the time interval, but this is a relative concept. It can be measured directly by using the performance evaluation weight within the government or indirectly by using the time interval of government response policies. Due to data availability limitations, this paper adopted an indirect measurement method; however, in the future, the two measurement methods could be combined for further detailed research.

### 6. Conclusions

First, a single variable is not a necessary condition for the formation of ULFCG. From the perspective of a single variable, the six factors of TOE cannot become the necessary conditions for ULCG alone, indicating that the individual element does not constitute the bottleneck of ULCG performance and further proving that multifactor joint is an effective way to drive ULCG.

Second, there are three different differentiated path patterns in the overall analysis. The overall path analysis indicated that the ULCG is composed of multiple synergies. There are five dynamic combination paths, which can be summarised into three categories—'big data- + market-led', 'big data-led', and 'market-led' ULCG modes—in terms of 'reaching the same goal through different paths' concerning the influence on ULCG performance.

Third, there is a clear substitution relationship between configuration S2 and configuration S3. The development level of big data in Configuration S2, as the core existence condition, and the marketisation level, as the edge-lacking condition, can be interchangeable with the development level of big data as the edge-lacking condition, and the marketisation level as the core existence condition in Configuration S3.

Fourth, there are significant differences in the paths of the eastern, central, and western regions of China. From the perspective of ULCG paths in the east, middle, and west, the core factors in the eastern region involve TOE, reflecting the complexity of governance paths in the eastern region. The central region is an organisation-oriented path with attention distribution as the core. The western region is a technology-led path with the development level of big data as the core. It can be found that there are obvious differences in the regional governance paths of eastern, central, and western China, further indicating that the factors-causing heterogeneity in ULCG levels is different.

**Author Contributions:** Conceptualization, Y.-D.L. and C.-L.Y.; methodology, Y.-H.Z.; software, J.-Q.B.; validation, Y.-D.L. and C.-L.Y.; formal analysis, C.-L.Y.; investigation, Y.-H.Z.; resources, Y.-D.L.; data curation, C.-L.Y.; writing—original draft preparation, Y.-D.L.; writing—review and editing, Y.-H.Z.; visualization, C.-L.Y.; supervision, J.-Q.B.; project administration, Y.-D.L. and Y.-H.Z. funding acquisition, Y.-D.L. All authors have read and agreed to the published version of the manuscript.

**Funding:** National Natural Science Foundation of China project "Research on decision optimization and coordination of low carbon supply chain considering equity cooperation under carbon regulation" (Project No. 72062023); Inner Mongolia natural science foundation project "Research on the influence mechanism of product sharing among consumers on enterprise operation decision-making" (Project No. 2019MS07026); Inner Mongolia natural science foundation project "Research on collaborative optimization of security inspection defence resource allocation of urban passenger transport hubs under normalized epidemic prevention"(Project No. 2022QN07003); Inner Mongolia Autonomous Region universities innovation team development plan support project "Big data and green governance" (NMGIRT 2202).

**Institutional Review Board Statement:** Not applicable.

**Informed Consent Statement:** Not applicable.

**Data Availability Statement:** Not applicable.

**Conflicts of Interest:** The authors declare no conflict of interest.

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
