# Peer review of "Analysing Multiple Paths of Urban Low-Carbon Governance: A Fuzzy-Set Qualitative Comparative Analysis Method Based on 35 Key Cities in China"

_sustainability, doi:10.3390/su15097613_

Round 1

Reviewer 1 Report

The presented article deals with the current topic while presenting mainly processed analyzes of data obtained on the basis of the statistical yearbook from 2020. It is therefore more of a review type than a research article.

1. As part of the analysis, 35 key municipalities were selected in eastern, central and western China, but there is lacking description of the methodology and criteria on the basis of which this selection would be justified.

2. The authors should justify why the statistical data of 31 key cities and four municipalities were selected from 2019 and were not compared with several years (for example, 2010, 2020, or last year 2022). The selected time period is not sufficient to assess the impact of various changing factors.

3. The abstract is divided into a non-standard structure, it needs to be reformulated.

4. The goal of the article is not clear from the introduction and should follow up on the basis of the overview of the development status within the topic. With an unclear goal, it is not possible to assess the appropriateness of the chosen methodology and the degree of achievement of the set goals. The conclusions are not clearly structured in order to assess the degree of novelty of the results achieved.

5. The graphic processing of configuration 1 in Figure 2 is unclear, which reduces the possibility of determining the position of the given city and its data.

6. According to the description in line 552 Fig. 4 contains detailed substitution effects, but in reality, any details are absent.

7. The authors refer to a literary source (Ragin, 1987) e.g. in line 298. However, that source is not listed in the reference list.

Based on the assessment, I recommend that the authors make extensive changes to the article.

Author Response

请查看附件

Reviewer 2 Report

Dear Authors,

thank you for interesting research and especially thank you for using fs/QSA as it is a really distinctive and little used tool. However, I have a few suggestions regarding your paper:

1. Could you elaborate the notion of marketisation and especially the marketisation level. It is not a very common term and maybe there are some translation issues. Also, please describe to a greater detail the calculation of marketisation level.

2. Please review your abstract. Currently it is very technical and does not reveal all the good aspects of the paper.

3. Please review Keywords, as it is the same issue as with the abstract.

4. Thank you for writing limitations and future research. 

5. It was very interesting to learn how you used TOE framework. I find it quite outdated, but you have used it in a very interesting and rich way.

Language quality is fine. However, there are a few minor typo issues.
